# Case Studies of US Research Information Management Practices

Proposal for VIVO 2021 conference

By Rebecca Bryant and Jan Fransen, 10 May 2021

## Proposal title

Case Studies of US Research Information Management Practices

## TL;DR

Learn about the status of research information management in the US

## Abstract (150-350 words)

This session will share the findings from a forthcoming OCLC Research report on *Research Information Management Practices in the United States* (http://oc.lc/us-rim-project), scheduled for early fall 2021. The report collects evidence from in-depth case studies of RIM practices at five US research universities:

- Penn State University
- Texas A&M University
- Virginia Tech
- UCLA
- University of Miami

The case studies represent open source, proprietary, and home grown RIM solutions at the five institutions and highlight the proliferation of use cases such as public portals, faculty activity reporting, and strategic reporting.

By synthesizing information from the five case studies, we offer a comprehensive definition of Research Information Management and also document the multiple use cases that proliferate in decentralized US research universities. We will also offer a new RIM System Framework, which describes the required and optional functional and technical elements that comprise the architecture of US RIM systems, regardless of use case. We believe that this framework will help demystify RIM infrastructure and also help practitioners better understand the array of campus stakeholders required for successful RIM implementation.

This research is based upon interviews with 39 participants engaged in RIM activities at the five case study institutions and builds upon the significant body of work on RIM practices already produced by OCLC Research (oc.lc/rim). We believe this research is of considerable utility to the university community, offering a more comprehensive and strategic view of RIM practices, along with recommendations for institutions.

We will conclude the presentation by demonstrating the value of the case studies and framework through examples pulled from the report's case studies.

## About the presenters

**Rebecca Bryant,** PhD, Senior Program Officer, OCLC
bryantr@oclc.org

Rebecca Bryant, PhD, serves as Senior Program Officer at OCLC Research where she leads investigations into scholarly communication topics such as research information management (RIM). Her OCLC publications include Research Information Management: Defining RIM and the Library's Role and Social Interoperability in Research Support: Cross-campus Partnerships and the University Research Enterprise.

Bryant began her career at the University of Illinois where she served as Assistant Dean in the Graduate College, Project Leader on the system-wide ERP project, and Project Manager for Researcher Information Services in the University Library. She was also formerly community director at ORCID.

**Jan Fransen**
Service Lead for Research Information Management Systems, University of Minnesota
fransen@umn.edu

Janet (Jan) Fransen is the Service Lead for Research Information Management Systems at University of Minnesota Libraries. In that role, she works across divisions and with campus partners to provide library systems and data that save researchers', students', and administrators' time and highlight the societal and technological impacts of the University's research. The most visible system in her portfolio is Experts@Minnesota, found at experts.umn.edu.

