# OpenReview forum: "Case Studies of US Research Information Management Practices"
_vivoconference.org/VIVO/2021/Conference_

### Official Review · Program_Chairs · 2021-06-01
**RIM framework and use cases of interest**

**Rating:** 8
**Confidence:** 4

**Review:**

The author is well-known in the area of RIM consideration and evaluation.  The work of OCLC in this area is of great value.

The RIM framework and use cases could be of interest to the VIVO community beyond institutions in the United States.

The US
setting, while familiar to many, has some peculiar features, in particular the nature of grant competition for research funding, the presence of large medical systems on university campuses, and the role of agricultural extension.  All five of these universities are major research universities.  Three are land grant.  Two have major medical systems on their campuses.  Understanding the nature of these institutions and the consequences for RIM systems at these institutions may be a challenge for non-US audience members.